# Integrated Design of the Vacuum and Safety Barrier between the Lithium and Test Systems of IFMIF-DONES

András Zsákai [1,*], Tamás Dézsi [1,2], András Korossy-Khayll [2], Imre Katona [1], Viktor Varga [2], Endre Kósa [2], Dénes Zoltán Oravecz [1], Santiago Becerril [3], Carlos Meléndez [4], Jesus Castellanos [5], Gioacchino Miccichè [6] and Angel Ibarra [7,8]

1   Centre for Energy Research (EK-CER), 1121 Budapest, Hungary
2   C3D Engineering Consultant Ltd., 1106 Budapest, Hungary
3   Vicerrectorado de Transferencia e Investigación, University of Granada (UGR), 18071 Granada, Spain
4   Esteyco S.A., 28036 Madrid, Spain
5   INAIA, Universidad de Castilla-La Mancha, 13071 Toledo, Spain
6   ENEA, Department of Fusion and Nuclear Safety Technology, C.R. Brasimone, 40032 Camugnano, Italy
7   Ciemat, Fusion Technology Division, 28040 Madrid, Spain
8   IFMIF-DONES España, 18071 Granada, Spain
*   Correspondence: zsakai.andras@ek-cer.hu

**Abstract:** The international fusion materials irradiation facility-DEMO-oriented neutron source (IFMIF-DONES) is a facility that is designed under the framework of the EU fusion roadmap. It is going to be an essential irradiation facility for testing and qualifying candidate materials under severe irradiation conditions of a neutron field having an energy spectrum like the one present in a fusion power reactor. The material specimens are irradiated in a containment structure named the test cell (TC), which is part of the test systems (TS). The TC also houses a part of the other major system (lithium system, LS), which provides the liquid lithium for the reaction through a piping system. At a point, the lithium piping needs to exit the TS, but the primary safety boundary must be continuous around these penetrations. Therefore, a special barrier, called the test systems–lithium systems interface cell (TLIC), has been developed around the piping system to provide a safety-approved and remotely maintainable vacuum boundary envelope. In this paper, the integrated design development of the TLIC is described, consisting of the design development according to the RCC-MRx code, the remote-handling (RH) needs, and the procedures and safety-related special needs of the design.

**Keywords:** IFMIF-DONES; TLIC; test cell; RCC-MRx; remote handling

## 1. Introduction

The main purpose of the international fusion materials irradiation facility-DEMO-oriented neutron source (IFMIF-DONES) facility is to provide a neutron source for irradiating small specimens and producing experimental data of material properties for the construction of the DEMO fusion power plant [1]. The plant will produce a 125 mA deuteron beam that is accelerated up to 40 MeV and shaped to have a nominal cross section in the range from 100 mm × 50 mm to 200 mm × 50 mm, which impinges on a liquid lithium curtain. The stripping reactions generate a large number of neutrons that interact with the materials' samples located immediately behind the lithium target, in the test modules.

The test cell (TC) of the DONES facility is a confined and well-shielded space where the strong irradiation reaction is created (Figure 1). The biological shielding of the TC mainly consists of several-meters-thick maintainable concrete walls called removable biological shielding blocks (RBSB) and maintainable shielding plugs (lower and upper shielding plugs: LSP and USP, respectively), and a stainless-steel liner called the test cell liner (TC liner). The confinement is closed at the top by the test cell cover plate (TCCP). The TC also

contains the test modules (not depicted on Figure 1), which house the material samples and also contain the target system (TSY) of the lithium systems (LS) [2], which provides a regulated lithium flow. The lithium piping exits at the bottom of the TC through the inlet and outlet pipe assembly (IPA and OPA, respectively) and connects to the primary lithium loop housed in the lithium room below the TC. The lithium loop mainly consists of the electromagnetic pump (EMP) circulating the liquid lithium, the heat exchanger connected to an oil-based secondary loop, and a dump tank that houses the lithium during maintenance and start-up and shut-down operations.

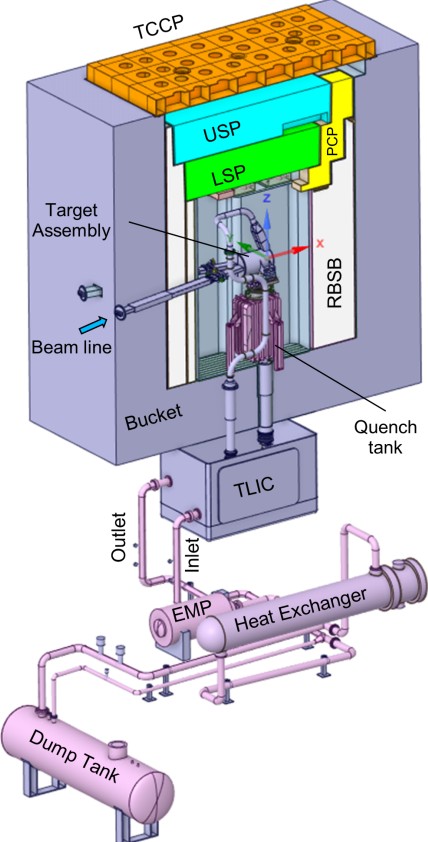

**Figure 1.** Test Cell and lithium loop of IFMIF-DONES with the space reservation of the TLIC.

The safety and vacuum barrier of the TC is provided by the TC liner generally and the TCCP at the top, but an additional confinement barrier is needed due to the IPA and OPA penetrations at the bottom. The initial idea to close the confinement at the bottom was to use a bellow construction at the ceiling of the lithium room, but this is not sufficient from a safety point of view due to the vulnerability of the bellow. Alternatively, a closed cap welded to the lithium pipe with a sealing at the bottom of the TC could serve as the boundary to separate the TC and lithium room; however this would mean that the sealing would be inside the TC liner in a high neutron streaming environment; therefore, it would need regular maintenance and enough space for remote-handling equipment to do the maintenance work. These two mentioned options have been rejected, and the concept of the test systems–lithium systems interface cell (TLIC) was selected [3] as the baseline, which is a box-like structure that envelopes the bellow construction connected to the inlet and outlet liquid lithium pipes; therefore, it serves as the vacuum and safety boundary between the TS and LS. The TLIC is considered as a direct extension of the TC liner welded to the the ceiling of the lithium room [4,5].

The TLIC needs to fulfil the following roles:

- Provide safety boundary—separating the test cell atmosphere from the lithium room;

- Maintain vacuum—the TC is under vacuum during normal operation, and the TLIC is an extension of the TC liner and is therefore part of the vacuum confinement;
- Maintainability—due to neutron streaming, remote handling is needed to maintain components of the TLIC; therefore, direct access from outside to the inside needs to be provided;
- Neutron shielding—if needed, neutron shielding should be provided inside or on the outside walls of the TLIC.

Moreover, the RCC-MRx design code [6] has been recently chosen to aid the design procedure of the TC liner; therefore, the TLIC (as an extension of the TC liner) should also follow the rules of the code.

## 2. Integrated Design of the TLIC

The optimization of the TLIC design is a complex problem as there are several fields of interest that superimpose requirements on the design and can be a strict design driver; therefore, an integrated design approach is a must to fulfil every need. In the following subsections, the different needs are elaborated in detail.

### 2.1. Remote-Handling Aspects

The TLIC is fixed to the ceiling inside the lithium room where the whole liquid lithium loop is situated. It is essentially a box structure with two sealed and screwed doors on the sides, and it envelopes the inlet and outlet pipe openings to the test cell. The lithium room is only accessible by remote-handling equipment during operation, and human access is still restricted during maintenance. Moreover, the activation of components inside the TLIC is over the hands-on limit according to neutronics calculation [7]; therefore, remote-handling equipment is needed to perform maintenance tasks on the lithium piping inside the TLIC and on the sealings and doors of the TLIC. The remote-handling tasks are going to be performed by a "strong arm–smart arm" concept (Figure 2), where a rigid strong arm is going to hold the components in place while an agile robotic arm is going to perform delicate tasks (screwing, cleaning, etc.). The RH equipment is transported by a customized automated guided vehicle (AGV) platform, and the doors are going to be lifted by a customized lifting frame fixed on the side of the platform. The main constraining requirements coming from RH on the TLIC are the size of the opening on the side walls, the needed space inside the TLIC, and the ability to reach the components inside on both sides to lower the moment forces on robotic equipment when handling lithium pipe segments during maintenance.

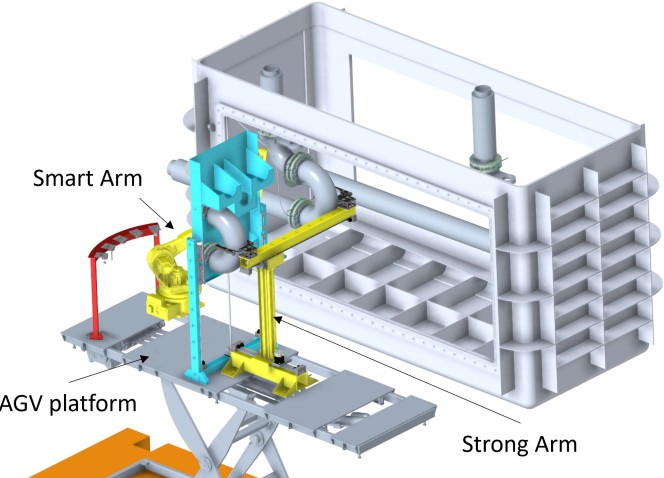

**Figure 2.** Remote-handling maintenance concept for the TLIC.

## 2.2. Safety Aspects

The TLIC is categorized as SIC-1 safety component [8], so the design must fulfil its function under any postulated initiating event in order to ensure compliance with the established dose limits. To aid this safety requirement, an appropriate nuclear design code should be used; therefore, the RCC-MRx code was chosen to support the design process even though it needs to be tailored to the Spanish legislation as the code itself is based partially on the French order on nuclear-pressure equipment (ESPN).

## 2.3. Atmosphere Separation

The atmosphere separation between the test cell and the TLIC is an open question; whether it is needed or not, the decision will be based on the remote-handling maintenance procedure of the whole test cell and lithium room. Currently, it is foreseen to have argon atmosphere inside the test cell and lithium room during maintenance, but the atmosphere in the TLIC could be different during final testing before start-up, which would imply the need for atmosphere separation to be able to maintain vacuum in the test cell. The current baseline includes a bellow-sealing concept to separate the atmospheres [9], which was not fit to serve as a safety boundary due to the vulnerability of the bellow.

## 2.4. Neutron Shielding

The design should also tentatively take into account if a neutron shielding layer is needed inside the TLIC due to the direct open lines through the lithium inlet and outlet pipings from the test cell into the lithium room. This layer can be formed by concrete blocks or by a polyethylene pellet layer inside the TLIC on the bottom.

## 3. RCC-MRx Categorization of the TLIC

The RCC-MRx categorization is based on the rules indicated by the European pressure equipment directive (PED) and the French pressure equipment directive (ESPN). The TLIC RCC-MRx codification follows the codification of the test cell liner as it is an integral part of it. Tentatively, the TC liner and therefore the TLIC is categorized as Class $N1_{R_x}$. The PED categorization is based on the volume and internal pressure of the vessel or on the governing pipe size and internal pressure of piping. A vacuum vessel in itself should not be regarded as a pressure equipment (PED Guideline A-57); however, in the case of the test cell liner, the active water cooling system (DN25, 6 bar) attached to the liner can be categorized as PED Cat-0 (PED R557-9-3-III and RCC-MRx REC 3221g) pressure equipment, and the liner and water cooling are regarded as a nuclear assembly. According to the ESPN code, equipment can be regarded as nuclear-pressure equipment if it is defined as pressure equipment by ESPN (aligned with PED); is used in a basic nuclear installation; directly ensures the containment of radioactive substances; and, in case of failure, leads to the release of activity above 370 MBq. A direct calculation has not yet been made, but due to the large containment size, it is assumed that the release of activity is above the threshold, and a preliminary N2 level category is assumed for the TC liner. Due to the fact that the TLIC is an extension of the TC liner, it can be regarded as part of the nuclear assembly, and the same categorization could apply to it as for the TC liner. Even if the categorization by ESPN and PED is chosen, which is the PED Cat 0. and ESPN N2 category , the RCC-MRx code still gives options to categorize the equipment (RCC-MRx REC 3231a). Tentatively, the $N1_{R_x}$ category is chosen for the TC liner and the TLIC is chosen too, due to their significance in DONES plant safety.

### Design Requirements of the TLIC

The goal of the analysis, besides reaching a consolidated detailed design, is to reduce the mass of the door as much as possible, to optimize the number of screws on the doors, and to optimize the stiffener layout.

The initial material selection for the TLIC components is SS316L, and preliminary simulations have been conducted to see the possibility of the weight reduction of the TLIC

doors. However, the optimum was still over 700 kg per door, which was considered too high for remote handling to handle; therefore, it was decided to use Aluminum 6061 T6 instead as a door material, which resulted in an optimum weight of 430 kg per door.

The analysis was considered elastic with negligible irradiation (maximum 0.02 dpa/20 years occur for SS316L [10]), negligible creep (temperatures below 450 °C for SS316L and below 100 °C for Al 6061 T6), and negligible fatigue load (the load cycles in the order of 20).

According to the $N1_{R_x}$ categorization of the TLIC, the following main rules need to be fulfilled.

For P-type damage (RCC-MRx RB 3251.112),

$$P_m \leq S_m; P_l \leq 1.5 * S_m; P_l + P_b \leq 1.5 * S_m \tag{1}$$

where $P_m$, $P_l$, $P_b$, and $S_m$ are the primary membrane stress, local membrane stress, bending stress, and allowable stress, respectively.

For S-type damage (RB 3261.1118), the following rules apply:

$$P_l + P_b + \Delta Q \leq 3 * S_m \tag{2}$$

where $\Delta Q$ is the secondary stress range.

The following loads must be considered according to the integral needs and in case of normal operation.

Primary loads (P):

- Maintain vacuum; 1 bar difference is assumed;
- Gravity load;
- Lithium pipe pressure: 7.8 bar lithium pressure.

Secondary loads (Q):

- Heat load of liquid lithium: 300 °C;
- Heat load at the lithium room ceiling connection (concrete embedment): 50 °C assumed on boundary;
- Heat map of TLIC approximated by steady state thermal calculation then implemented on the structural model;
- Volumetric heating was considered negligible for this analysis.

## 4. Detailed Design Analysis and Optimization

The main sizes of the TLIC were determined by the components covered inside and the needed space for RH equipment manoeuvring. The analysis was carried out on a simplified shell model of the TLIC, and mesh independence shows a few percent difference between the coarse, rough, and fine meshes. An optimization has been carried out on several components and features of the TLIC, e.g., on the main body thickness size, on the door geometry including flat faced door design, on the varied elevated inner structure of the door, and on the main body stiffening structure height and positioning. An optimized layout emerged, which consists of varying body thicknesses of 10–30 mm, a 15 mm thick, and 200 mm high stiffener layout on the outside and inside of the TLIC body; a fixed welded connection to the lithium piping; low-mass doors; and an optimized number of screws on doors .

The deformation field of the model shows a maximum deflection of 3.7 mm in case of primary loads, with a maximum deflection of 2.5 mm in the TLIC doors. The primary membrane stress of the TLIC body is below 147 MPa at 150 °C or lower temperatures and below 115 MPa at 300 °C, which are the allowable stresses for SS316L (RCC-MRx Table A3.1S.41). The primary membrane and bending stresses of the TLIC body are below the allowable limit (220 MPa below 150 °C and 172.5 MPa at 300 °C) throughout the whole area (Figure 3).

The TLIC door is modeled in a hybrid way; the base is a solid model, while the rest is made up of shell elements (Figure 4). The primary membrane and bending stresses

can be directly shown on a shell model without using a stress-classification method. The results are shown in the middle (membrane) or on the top/bottom sides of the shell element (membrane+bending). The top plate of the assembly is omitted from the analysis as it is decoupled from the rest of the model due to the direct concrete embedment of the TLIC body to the ceiling of the lithium room. The solid-shell connection is modeled away from welding lines to correctly capture stresses in the most vulnerable cross sections. Generally, the primary stresses and secondary stress range of the doors satisfy the given criteria; the largest stresses occurred in the door base close to the top of the TLIC, so this area was chosen for further analysis using stress classification lines (SCL) along the welded cross section and along a line through the base itself. The stresses in the welded cross-section are calculated using a weld factor of J = 0.6 (RCC-MRx table A9.J2A.43). The results show that the stresses are below the limit, and therefore the design is considered suitable (Figure 4) and Table 1.

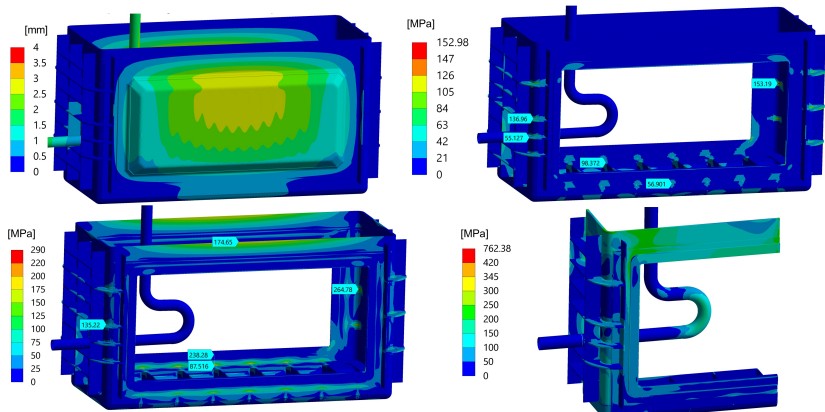

**Figure 3. Top left**: Deformation of the TLIC and the TLIC door due to primary stresses. **Top right**: primary membrane stresses of the main body; **bottom left**: primary membrane + bending stresses; and **bottom right**: primary membrane + bending stresses + secondary stress range.

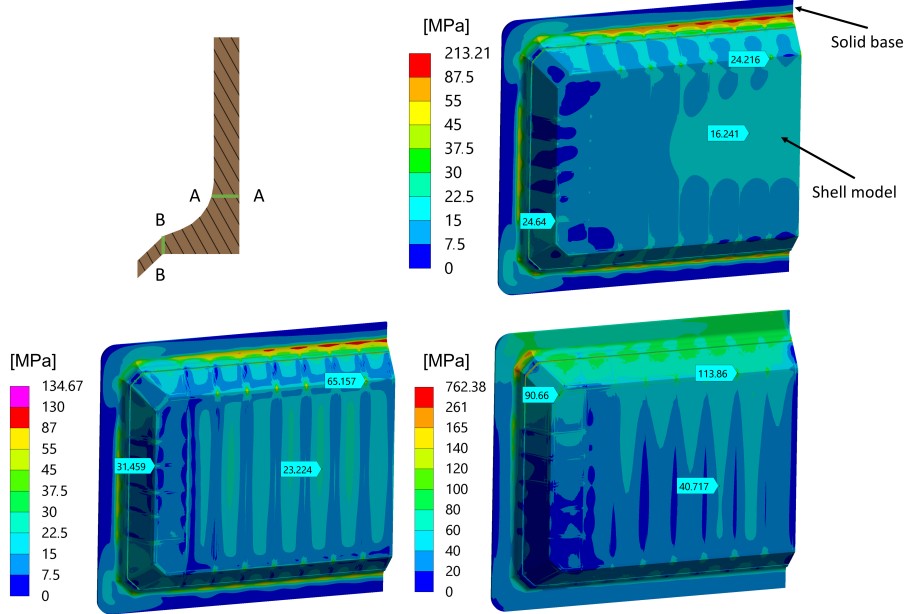

**Figure 4.** P-type and S-type damage evaluation of the TLIC door. **Top left**: SCL line definition in the solid door base; **top right**: primary membrane stresses; **bottom left**: primary membrane + bending stresses; and **bottom right**: primary membrane + bending stresses + secondary stress range.

**Table 1.** Primary stresses in the SCL lines of the TLIC door (Al 6061 T6).

| SCL Line | $P_M$ (MPa) | $S_M$ (MPa) | $P_L + P_B$ (MPa) | $1.5 \times S_M$ (MPa) |
|---|---|---|---|---|
| Through base thickness (A-A) | 18.4 | 87 | 94.7 | 130.5 |
| Through welding of inclined plate to base (B-B) | 27.6 | 55 | 53.3 | 82.5 |

Primary membrane and bending stresses + the secondary stress range in the TLIC body are below the allowable limit (420 MPa below 150 °C and 345 MPa at 300 °C for SS316L). The TLIC door shows a peak load in the corner of the base, which was chosen to be further analyzed. Two SCL lines are drawn (Figure 4), one alongside the base thickness and one alongside the inclined welding connection. The results show that the secondary stresses are below the allowable limit, and therefore the design is considered suitable for secondary stresses too (Table 2).

**Table 2.** Primary membrane + primary bending + secondary stresses in the SCL lines of the TLIC door (Al 6061 T6).

| SCL Line | $Max\overline{P_L + P_B} + \overline{\Delta Q}$ (MPa) | $3 \times S_M$ (MPa) |
|---|---|---|
| Through base thickness (A-A) | 101.3 | 261 |
| Through welding of inclined plate to base (B-B) | 82.5 | 165 |

A preliminary linear elastic buckling analysis on a perfect nominal geometry was also carried out on the model to capture the most harmful buckling modes. The first mode occurs on the aluminum TLIC door with a load multiplier of 6.11 (Figure 5). The design is considered suitable for buckling as the first mode is not going to be a dominant failure mode due to the high load multiplier; a more detailed analysis will be carried out in a future step.

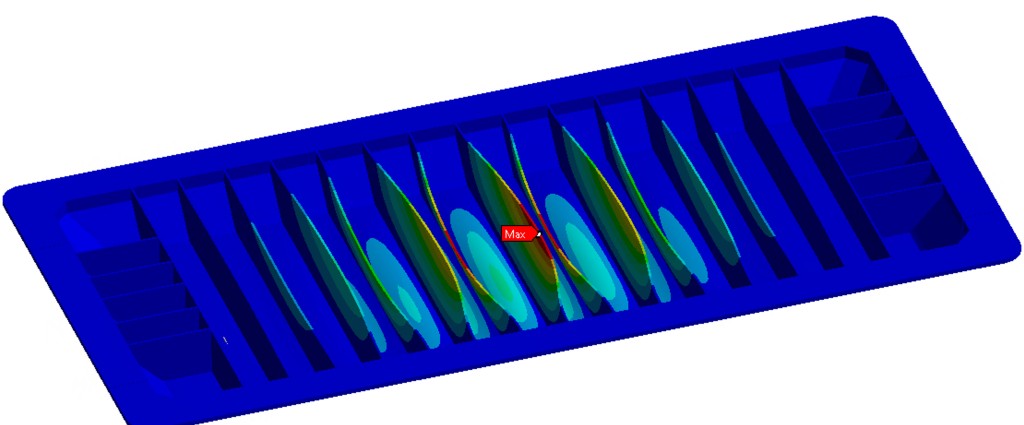

**Figure 5.** First buckling mode of the model.

The bolting and sealing of the TLIC doors have been optimized to have a low number of bolts while still maintaining vacuum conditions. RCC-MRx rules A6.2320 restrain the positioning of bolts to manufactured edges and to each other; as a conservative approach, the design is adapted to these rules, resulting in an optimized version that still has 78 remote-handling-compatible M20 captive bolts.

The sealing design includes a double-layered 1/4″ O-ring sealing outline with double dovetail grooves to maintain the position of the sealings inside the TLIC doors during maintenance. An analysis in the case of the application of vacuum + lithium pressure and in the case of normal operation (heat loads) was carried out to see the resulting gap alongside the sealings. The allowable gap size is estimated as 10% of the O-ring diameter

as a good engineering practice. The resultant gaps are below the threshold; therefore, the design is considered suitable (Table 3 and Figure 6).

**Table 3.** Gap result evaluation of the TLIC door sealing design.

| Loads | Max. Gap Size (mm) | Allowable Gap Size (mm) |
|---|---|---|
| Primary load case | 0.0064 | 0.63 |
| Primary + Secondary load case | 0.1226 | 0.63 |

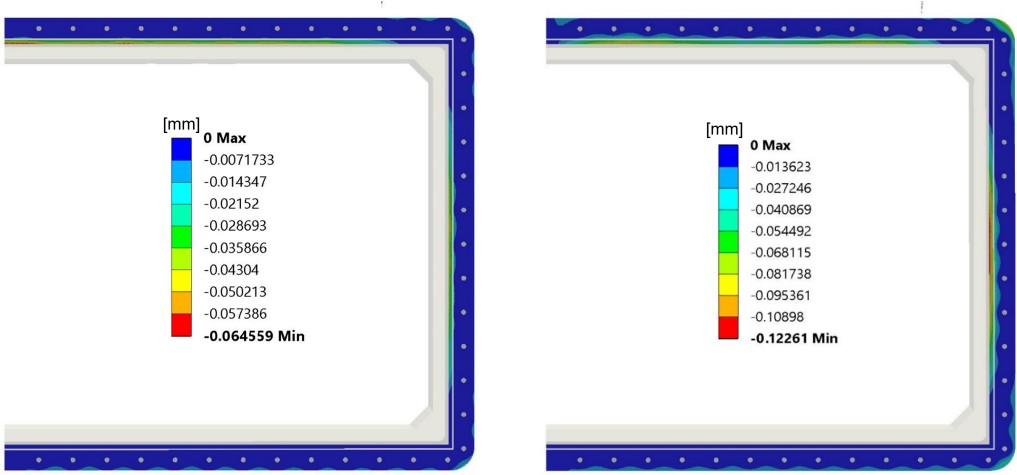

**Figure 6.** Gap sizes along the base of the TLIC door. **Left**: resulting gap in case of deadweight + vacuum + lithium pipe pressure loads applied; **right**: resulting gap in case of normal operation.

## 5. Results

The final version of the TLIC detailed design fulfils the design criteria imposed by the integral areas and by the RCC-MRx code, both for P-type and S-type damage and for buckling. The final model also includes minor details such as remote-handling positioners and handles on the door (Figure 7).

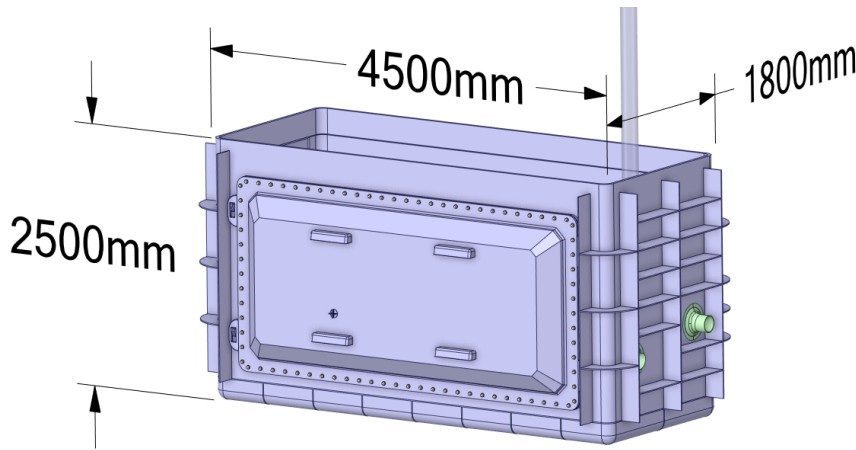

**Figure 7.** Detailed design model of TLIC.

## 6. Conclusions

A preliminary box-like structure has been proposed to provide safety and a vacuum boundary between the TS and LS enveloping the lithium inlet and outlet pipes. Several areas (safety, RH, TS, LS, etc.) impose requirements on this bounding box; therefore, an integrated design analysis had to be carried out to propose a detailed design basis. In order to aid the design process, the application of the RCC-MRx code has been introduced and

elaborated on. The design has been confirmed to withstand normal operation loads and that it can fulfil requirements from all other areas. Further analysis could include off-normal event evaluation and possible accident-scenario analysis.

**Author Contributions:** Writing—orgiinal draft, review & editing, A.Z. and T.D.; Resources, A.K.-K.; Formal Analysis, A.Z. and I.K.; Methodolgy, V.V. and E.K.; Visualization, D.Z.O.; Validation, S.B. and C.M.; Supervision, J.C., G.M. and A.I. All authors have read and agreed to the published version of the manuscript.

**Funding:** This work has been carried out within the framework of the EUROfusion Consortium, funded by the European Union via the Euratom Research and Training Programme (Grant Agreement No 101052200—EUROfusion).

**Data Availability Statement:** Not applicable.

**Acknowledgments:** This work has been carried out within the framework of the EUROfusion Consortium, funded by the European Union via the Euratom Research and Training Programme (Grant Agreement No 101052200—EUROfusion). The views and opinions expressed are, however, those of the author(s) only and do not necessarily reflect those of the European Union or the European Commission. Neither the European Union nor the European Commission can be held responsible for them.

**Conflicts of Interest:** The authors declare no conflict of interest.

## Abbreviations

The following abbreviations are used in this manuscript:

| | |
|---|---|
| AGV | automated guided vehicle |
| EMP | electromagnetic pump |
| ESPN | French order on nuclear-pressure equipment |
| IFMIF-DONES | international fusion materials irradiation facility-DEMO-Oriented Neutron Source |
| IPA | inlet plug assembly |
| LS | lithium systems |
| LSP | lower shielding plug |
| OPA | outlet plug assembly |
| PED | European pressure equipment Directive |
| RBSB | removable biological shielding BLocks |
| RH | remote handling |
| TC | test cell |
| TLIC | test cell–lithium systems interface cell |
| TCCP | test cell cover plate |
| TS | test systems |
| TSY | target system |
| USP | upper shielding plug |

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
