# Peer review of "Integrated Design of the Vacuum and Safety Barrier between the Lithium and Test Systems of IFMIF-DONES"

_jne, doi:10.3390/jne4010004_

Round 1

Reviewer 1 Report

The paper presents an interesting general overview of the design of the TLIC safety envelope of the IFMIF-DONES facility. In my opinion, this is a good contribution which is worthy to be published provided that minor revisions are made and explanations are given about some unclear points, as proposed in the following:  

-Abstract: It is not usual to have references in the abstract. In principle they should be avoided as the abstract should be kept self-standing. So please, consider moving them to the main text

-Introduction: A few lines at the beginning of Sect. 1 would be appropriate to frame the subject (in the same way as done in the abstract). In particular, it would be helpful to clearly explain why there is the need of a TLIC (it seems that the major reason is to create a safety boundary around the bellows of the Li pipes but this is not clearly stated in the Introduction)

-line 19: "Another option could be...": is this a different issue or an alternative solution to the problem of bellow vulnerability mentioned earlier? Please, make it clearer

-line 24: acronym TLIC was never defined before in the Introduction

-line 30: "Maintain vacuum": what is the reason to keep the TLIC (which is a volume outside of the TC) under vacuum? Please, briefly justify

-line 51: "human means" -> "human beings"?

-line 67: Is ESPN directive applicable given that the TLIC is under vacuum (and not under pressure)? Please, justify

-line 73: "acceptance testing": which kind of acceptance testing is referred to in particular? And why it would need a separate atmosphere? Please, add some more detail about this point 

-line 81: what does "PE" stand for?

-lines 83-86: why is the TC liner (and consequently the TLIC) classified as N1Rx? Does this really come on the basis of PED/ESPN classification as stated at the beginning of the Section? And in that case, what is the identified PED/ESPN category? Please, add some complementary information on these aspects

-line 86: "Tentatively, the Test Cell and..." -> "Tentatively, the Test Cell liner and..."?   

-Figures 3 and 4: it is not clear to me what the shown stress fields represent. In fact, they are referred to as "Primary membrane stresses", "Primary membrane+bending stresses" etc... but according to RCC-MRx terminology these quantities can only be defined in relation to predefined "supporting line segments" (or "stress linearization paths") while in Figs. 3 and 4 it looks like they are 3D distributed (continuous) fields. Otherwise, if they are actually calculated over stress linearization paths, could you please indicate where these paths are located?

-Figure 4: Indication (by arrows and corresponding text labels) of the solid "base" part, of the "shell" part and of the welds regions as well as of the "SCL lines" used for the calculations reported in Tabs. 1 and 2 would be very helpful for the reader in my opinion

-line 142: "stress classification lines (SCL)" -> "stress linearization lines (SLL)"?       

-line 170: "(Table 3 and Figure (Figure 6))"  -> "(Table 3 and Figure 6)"

-Table 3: why are the results kept separated for Primary and Secondary loads? Shouldn't they be considered in combination?

-Figure 6: which cases do the left-hand and the righ-hand pictures refer to? Please, indicate it in the caption 

Author Response

Thank you for the extensive review and comments added to the manuscript! I answered all questions and comments and corrected the manuscript according to them. You can find my answeres below in Italic.

-AbstractReferences removed from Abstract and moved to enhanced introduction section.

-Introduction: Introduction section enhanced and should give a clear understanding now.

-line 19: "Another option could be...": is this a different issue or an alternative solution to the problem of bellow vulnerability mentioned earlier? Please, make it clearer

Yes, it would also serve as the safety and vacuum boundary in this case, therefore the bellow construction would be safe to use. Wording is corrected.

-line 24: acronym TLIC was never defined before in the Introduction

Acronyms corrected generally in the manuscript and an abbreviation section added at the end of the manuscript.

-line 30: "Maintain vacuum": what is the reason to keep the TLIC (which is a volume outside of the TC) under vacuum? Please, briefly justify

The Test Cell is under vacuum during normal operation and the TLIC is an extension of the TC, therefore it is also serving as the safety and vacuum boundary, that is why it must be under vacuum too. Sentence corrected accordingly in the paper.

-line 51: "human means" -> "human beings"?

Yes, sentence corrected to: The lithium room is only accessible by remote handling equipment during operation and human access is still restricted during maintenance.

-line 67: Is ESPN directive applicable given that the TLIC is under vacuum (and not under pressure)? Please, justify

If the TLIC would be a separate vacuum vessel standing in itself, then the PED and ESPN categorization would not stand as the PED clearly defines that an equipment is not regarded as a pressure equipment if it is under vacuum, however the TLIC, the Test Cell liner and the liner cooling form a nuclear assembly and the cooling circuit is regarded as a pressure equipment therefore the same rules should apply to the nuclear assembly. Initially, this reasoning sought to be included in the cited paper about the test cell liner design, however that will be realeased in another issue due to other reasons, therefore the RCC-MRx section is filled in with more information.

-line 73: "acceptance testing": which kind of acceptance testing is referred to in particular? And why it would need a separate atmosphere? Please, add some more detail about this point 

Sentence elaborated, the meaning of acceptance testing is the final testing of the equipment at the end of a maintenance procedure (flange test, seal test, sensor testing, etc.). The separate atmosphere would be needed in case the TLIC final testing can only be done in case the Test Cell is already under vacuum.

-line 81: what does "PE" stand for?

stands for Polyethylene, abbreviation omitted from text.

-lines 83-86: why is the TC liner (and consequently the TLIC) classified as N1Rx? Does this really come on the basis of PED/ESPN classification as stated at the beginning of the Section? And in that case, what is the identified PED/ESPN category? Please, add some complementary information on these aspects

PED/ESPN categorization details and classification justification added to RCC-MRx section.

-line 86: "Tentatively, the Test Cell and..." -> "Tentatively, the Test Cell liner and..."? 

Yes and corrected.

-Figures 3 and 4: it is not clear to me what the shown stress fields represent. In fact, they are referred to as "Primary membrane stresses", "Primary membrane+bending stresses" etc... but according to RCC-MRx terminology these quantities can only be defined in relation to predefined "supporting line segments" (or "stress linearization paths") while in Figs. 3 and 4 it looks like they are 3D distributed (continuous) fields. Otherwise, if they are actually calculated over stress linearization paths, could you please indicate where these paths are located?

Yes, that is correct in case of an analysis of an actual 3D model (thickness also modeled), however the model has been simplified mainly to a shell model (only exception is the door base in which case, stresses have been calculated along SCL lines shown in tables 1 and 2). In case of the shell model, ansys allows to print results of the shell elements in the middle (giving directly the membrane stress as result) and on the top/bottom of the element (giving directly the membrane+bending stress). However, it is true thart this is not elaborated in the, therefore it is corrected.

-Figure 4: Indication (by arrows and corresponding text labels) of the solid "base" part, of the "shell" part and of the welds regions as well as of the "SCL lines" used for the calculations reported in Tabs. 1 and 2 would be very helpful for the reader in my opinion

Depictions on figures included

-line 142: "stress classification lines (SCL)" -> "stress linearization lines (SLL)"?       

The RCC-MRx code does not define the naming of the line, however the common term used generally is stress classification line (SCL), thus it is left unchanged.

-line 170: "(Table 3 and Figure (Figure 6))"  -> "(Table 3 and Figure 6)"

Corrected.

-Table 3: why are the results kept separated for Primary and Secondary loads? Shouldn't they be considered in combination?

The wording is misleading in this case, the two cases show the results under Primary loads and Primary+Secondary loads. Corrected in the table.

-Figure 6: which cases do the left-hand and the righ-hand pictures refer to? Please, indicate it in the caption 

Caption corrected to indicate which picture belongs to which case.

Reviewer 2 Report

SUMMARY

Safe design and licensing of a fusion reactor demand the understanding degradation of materials under neutrons bombardment. The plasma facing components shall  withstand damage caused by the severe working conditions to which they are subjected during reactor operation.

In this context, IFMIF-DONES (International Fusion Materials Irradiation Facility-DEMO-Oriented NEutron Source) is a single-sited novel Research Infrastructure for testing, validation and qualification of the materials to be used in a fusion reactor.

In this context, the paper performs studies about Test Systems Lithium Systems Interface Cell (TLIC) barrier. A design has been developed to provide a safety approved and remotely maintainable vacuum boundary envelope.

The paper is of interest to readers of J. Nucl. Eng.

COMMENTS

In the following the weaker aspects of this paper are reported:

·         Section about  IFMIF-DONES Design  should be added (in which context is the described research activities?)

·         Target System that  includes the components located inside the Test Cell should be described in more detail

·         It is not clear if the main of this paper is a consolidated preliminary engineering design baseline for the IFMIF-DONES facility.

·         In my opinion, this paper lacks of novelty and scientific innovation (or at least Authors  don't highlight  this aspect in the paper text).

·         RCC-MRx code should ne describe in a dedicated section. This can help the reader to better understand the analyses and results. 

SPECIFIC COMMENTS

·         Abstract: Avoid formal citations in an abstract if at all possible. The reason is that if the reader  read paper abstract in isolation would not have an easy means to look up the citation. This can be done in a most suitable section (e.g. “Introduction”)

·         Introduction: As required by “ Instructions for Authors” of  J. Nucl. Eng.,  Introduction should define both the purpose of the work and its significance, including specific hypotheses being tested. The current state of the research field should be reviewed carefully and key publications cited.

·         Figure 1. Test Cell and Lithium Loop of IFMIF-DONES with the space reservation of the TLIC. Acronyms and abbreviations are not described in the paper.

·         Section 2.1. Remote Handling aspects: It should be contain further information useful for the reader to understand the system (same considerations should be applied in subsequent sections)

·         Section 3 line 99” According to the N1Rx categorization of the TLIC, the following main rules need to be fulfilled.” It is  fundamental that Authors give as much information as possible to help some way towards a better understanding of the text by the  reader.

·         Fig. 3 and 4 should be improved (in legend, color scale bar and text  are not clear)

·         I suggest to add in the paper a section “Conclusion”. A well-written conclusion provides Authors with important opportunities to demonstrate to the reader your understanding of the research problem.

Author Response

Thank you for the extensive review and comments added to the manuscript! I answered all questions and comments and corrected the manuscript according to them. You can find my answeres below in Italic.

COMMENTS

In the following the weaker aspects of this paper are reported:

  • Section about  IFMIF-DONES Design  should be added (in which context is the described research activities?)
  • Target System that  includes the components located inside the Test Cell should be described in more detail
  • It is not clear if the main of this paper is a consolidated preliminary engineering design baseline for the IFMIF-DONES facility.
  • In my opinion, this paper lacks of novelty and scientific innovation (or at least Authors  don't highlight  this aspect in the paper text).
  • RCC-MRx code should ne describe in a dedicated section. This can help the reader to better understand the analyses and results.

Introduction updated to include short description on IFMIF-DONES and its purpose, and Test Cell and Lithium components.

In fact, it is a preliminary engineering design baseline, text modified to include this.

RCC-MRx section added.

SPECIFIC COMMENTS

  • Abstract: Avoid formal citations in an abstract if at all possible. The reason is that if the reader  read paper abstract in isolation would not have an easy means to look up the citation. This can be done in a most suitable section (e.g. “Introduction”)

References removed from Abstract and moved to enhanced introduction section.

  • Introduction: As required by “ Instructions for Authors” of  J. Nucl. Eng.,  Introduction should define both the purpose of the work and its significance, including specific hypotheses being tested. The current state of the research field should be reviewed carefully and key publications cited.

Introduction section enhanced and should give a much clearer understanding.

  • Figure 1. Test Cell and Lithium Loop of IFMIF-DONES with the space reservation of the TLIC. Acronyms and abbreviations are not described in the paper.

Introduction section includes short description on components and describe abbreviations too. An abbreviation sectino at the end of teh manuscript is added too.

  • Section 2.1. Remote Handling aspects: It should be contain further information useful for the reader to understand the system (same considerationsshould be applied in subsequent sections)

Added information to section for better understanding the requirements imposed on the design.

  • Section 3 line 99” According to the N1Rx categorization of the TLIC, the following main rules need to be fulfilled.” It is  fundamental that Authors give as much information as possible to help some way towards a better understanding of the text by the  reader.

P-type and S-type damages elaborated in the text.

  • Fig. 3 and 4 should be improved (in legend, color scale bar and text  are not clear)

Figures updated.

  • I suggest to add in the paper a section “Conclusion”. A well-written conclusion provides Authors with important opportunities to demonstrate to the reader your understanding of the research problem.

A conclusion section is added to the manuscript to detail further.

Reviewer 3 Report

The paper deals with the description of the design of special barrier between two key subsystems of IFMIF-DONES design: Lithium and Test systems. The work is of important significance for safe operation of the planned installation.

Several comments and suggestion for improving the paper are given below:

1. In the Introduction there is no short general information about IFMIF-DONES objectives and main ideas. This is in the abstract, but it seems that should be included in the Introduction, in which in principle the authors should explain the reader why this work has been performed, what is supposed to achieve and what has been already done.

2. If the authors could add short info about RCC-MRx code (first mentioned in the Introduction) and why this code has been chosen, it would be convenient for the reader.

3. Point 2.3 concerns atmosphere separation. It is described as an open question. However - what would be in case of unexpected contamination - is there any procedure ?

4. In lines 101 and 104 (point 3.1) two types of damages are mentioned: P-type and Q-type. It would be good to include short information on how actually these damages are defined.

5. Line 120: was -> were (or sizes -> size)

6. Point 4. Detailed analysis and optimization. While in previous section design requirements are clearly defined, it is not fully clear what the authors understand by optimization. For example, in line 123 there is info about several rounds of simulations - what kind of simulations, what was the aim of these simulations ? I suggest to add information about the steps performed to obtain optimized model, for example in the form of a sort of algorithm.

7. Tables 1, 2, 3 are wider that the text.

8. Final section about results is very short - I think, that at least it should be mentioned what was the purpose of the work and what has been achieved.

Author Response

Thank you for the extensive review and comments added to the manuscript! I answered all questions and comments and corrected the manuscript according to them. You can find my answeres below in Italic.

Several comments and suggestion for improving the paper are given below:

  1. In the Introduction there is no short general information about IFMIF-DONES objectives and main ideas. This is in the abstract, but it seems that should be included in the Introduction, in which in principle the authors should explain the reader why this work has been performed, what is supposed to achieve and what has been already done.

Introduction updated to include short description on IFMIF-DONES, its components and their purpose.

  1. If the authors could add short info about RCC-MRx code (first mentioned in the Introduction) and why this code has been chosen, it would be convenient for the reader.

PED/ESPN categorization details added to RCC-MRx section.

  1. Point 2.3 concerns atmosphere separation. It is described as an open question. However - what would be in case of unexpected contamination - is there any procedure?

Procedures are under investigation as of now, in general a breakage should not be a problem in itself between TLIC and TC during normal operation, the problematic situation could be when the TLIC is under maintenance and the atmosphere is separated from the TC which is already under vacuum, because acceptance tests are needed ont he lithium piping flanges inside the TLIC before screwing of the TLIC doors.

  1. In lines 101 and 104 (point 3.1) two types of damages are mentioned: P-type and Q-type. It would be good to include short information on how actually these damages are defined.

P-type and S-type damages elaborated in the text.

  1. Line 120: was -> were (or sizes -> size)

Corrected.

  1. Point 4. Detailed analysis and optimization. While in previous section design requirements are clearly defined, it is not fully clear what the authors understand by optimization. For example, in line 123 there is info about several rounds of simulations - what kind of simulations, what was the aim of these simulations ? I suggest to add information about the steps performed to obtain optimized model, for example in the form of a sort of algorithm.

Further detailing added to the text on which components optimization has been conducted.

  1. Tables 1, 2, 3 are wider that the text.

This is a recommended usage of tables according to the MDPI template.

  1. Final section about results is very short - I think, that at least it should be mentioned what was the purpose of the work and what has been achieved.

A conclusion section is added to the manuscript to detail further.

Reviewer 4 Report

This paper described the design of the Test Systems Lithium Systems Interface Cell of IFMIF-DONES. Since it is important to inform the current design of the IFMIF-DONES components to the fusion community, the reviewer thinks this paper, which is presented in SOFT2022, is suitable for publication in Journal of Nuclear Engineering. The reviewer thinks that the technical description is appropriate but the manuscript should be modified in its expression style for readers’ better understanding. The reviewer would like to recommend the manuscript to be published after revising according to the comments below.

Page 1, abstract

According to the reviewer’s knowledge, the abstract should not refer to references.

Page 1, Introduction

According to the reviewer’s knowledge, the main text should be understandable without reading abstract. However, the introduction does not contain the background of the research. The beginning parts of the abstract should be included in the Introduction.

Page 1, Introduction

It is difficult to understand the first three sentences without referring a suitable figure. It is better to refer Fig. 1 earlier.

Page 2, Fig. 1

It is better to indicate which parts are the “test cell” and “lithium room” with its ceiling in the figure.

The abbreviations such as “TCCP”, “USP”, “LSP”, “RBSB” and “EMP” should be spelled out.

Page 4, line 107

What is “CAT-1”?

Page 5, line 126

Please indicate the “door” in Fig. 3.

Page 6, line 153

The “(Figure (Table 2)” is strange.

Page 7, line 170

The “(Table 3 and Figure (Figure 6))” is strange.

Author Response

Thank you for the extensive review and comments added to the manuscript! I answered all questions and comments and corrected the manuscript according to them. You can find my answeres below in Italic.

Page 1, abstract

According to the reviewer’s knowledge, the abstract should not refer to references.

References removed from Abstract and moved to enhanced introduction section.

Page 1, Introduction

According to the reviewer’s knowledge, the main text should be understandable without reading abstract. However, the introduction does not contain the background of the research. The beginning parts of the abstract should be included in the Introduction.

Introduction section enhanced and should give a clear understanding now.

Page 1, Introduction

It is difficult to understand the first three sentences without referring a suitable figure. It is better to refer Fig. 1 earlier.

Introduction updated and reference placed earlier.

Page 2, Fig. 1

It is better to indicate which parts are the “test cell” and “lithium room” with its ceiling in the figure.

The abbreviations such as “TCCP”, “USP”, “LSP”, “RBSB” and “EMP” should be spelled out.

Abbreviations are now explained and an abbreviation section added at the end of the document.

Page 4, line 107

What is “CAT-1”?

It would refer to loads categorized as main loads in case of normal operation (in RCC-MRx), however for understanding the text this is not needed, so removed.

Page 5, line 126

Please indicate the “door” in Fig. 3.

Door indicated in caption. The TLIC door is depicted also separately on Figure 4.

Page 6, line 153

The “(Figure (Table 2)” is strange.

Corrected.

Page 7, line 170

The “(Table 3 and Figure (Figure 6))” is strange.

Corrected.

Round 2

Reviewer 2 Report

no comments from my side